# Retinoic Acid Signaling Is Associated with Cell Proliferation, Muscle Cell Dedifferentiation, and Overall Rudiment Size during Intestinal Regeneration in the Sea Cucumber, *Holothuria glaberrima*

**DOI:** 10.3390/biom9120873

**Published:** 2019-12-13

**Authors:** Jorge Viera-Vera, José E. García-Arrarás

**Affiliations:** Biology Department, University of Puerto Rico, Río Piedras, PR 00931, USA; jorge.viera@upr.edu

**Keywords:** retinoic acid signaling, regeneration, sea cucumber, cell dedifferentiation, cell division

## Abstract

Almost every organism has the ability of repairing damaged tissues or replacing lost and worn out body parts, nevertheless the degree of the response substantially differs between each species. Adult sea cucumbers from the *Holothuria glaberrima* species can eviscerate various organs and the intestinal system is the first one to regenerate. This process involves the formation of a blastema-like structure that derives from the torn mesentery edges by the intervention of specific cellular processes (e.g., cell dedifferentiation and division). Still, the genetic networks controlling the regenerative response in this model system are just starting to be unraveled. In this work we examined if and how the retinoic acid (RA) signaling pathway is involved in the regenerative response of this deuterostome. We first identified and characterized the holothurian orthologs for short chain dehydrogenase/reductase 7 (SDR7) and aldehyde dehydrogenase family 8A1 (ALDH8A1), two enzymes respectively associated with retinaldehyde and RA anabolism. We then showed that the SDR7 transcript was differentially expressed during specific stages of intestinal regeneration while ALDH8A1 did not show significant differences in regenerating tissues when compared to those of normal (non-eviscerated) organisms. Finally, we investigated the consequences of modulating RA signaling during intestinal regeneration using pharmacological tools. We showed that application of an inhibitor (citral) of the enzyme synthesizing RA or a retinoic acid receptor (RAR) antagonist (LE135) resulted in organisms with a significantly smaller intestinal rudiment when compared to those treated with DMSO (vehicle). The two inhibitors caused a reduction in cell division and cell dedifferentiation in the new regenerate when compared to organisms treated with DMSO. Results of treatment with tazarotene (an RAR agonist) were not significantly different from the control. Taken together, these results suggest that the RA signaling pathway is regulating the cellular processes that are crucial for intestinal regeneration to occur. Thus, RA might be playing a role in echinoderm regeneration that is similar to what has been described in other animal systems.

## 1. Introduction

Regeneration studies have been closely associated with studies of embryological development. Many of these studies are based on the assumption that the formation of a tissue or an organ during regeneration will use mechanisms similar to those employed during embryonic organ formation. Thus, the signaling molecules that are known to regulate developmental processes have been the targets of many regenerative studies. One of these signaling molecules is retinoic acid (RA). For over 50 years, clinical and basic researchers have revealed that RA regulates multiple processes during embryonic development, such as axial and regional patterning, organogenesis, limb formation, and neurogenesis [1]. Experimental evidence has shown that RA influences RNA and protein synthesis by directly modulating transcription of specific genes [2]. Furthermore, it has been shown that RA signaling coordinates processes related to tissue homeostasis and organ regeneration through the regulation of multiple cellular processes, such as apoptosis, axial patterning, dedifferentiation, or cell cycle progression [3,4,5,6,7].

RA is a small lipophilic metabolite derived from retinoids (vitamin A and derivatives) or carotenoids that cannot be synthesized de novo by vertebrates. In adult animals, it is absorbed from their dietary regime as retinyl esters (RE), retinol (ROH—vitamin A), and as small amounts of RA when it originates from animal food sources or mainly as β-carotene from plant-based ones (vegetables and fruits) [8,9]. The biochemical conversion of retinol into retinoic acid arises from two consecutive enzymatic steps. In the first one, members of the medium-chain dehydrogenase/reductase family (e.g., alcohol dehydrogenases—ADHs) or from the short-chain dehydrogenase/reductase family (e.g., retinol dehydrogenases—RDHs) catalyze the oxidation of ROH into retinaldehyde. In the second one, an aldehyde dehydrogenase (e.g., retinaldehyde dehydrogenase—RALDHs) transforms retinaldehyde into RA, the active metabolite of vitamin A [4,10,11,12]. Alternatively, β-carotene is cleaved by β-carotene oxygenase directly into retinaldehyde and used to synthesize RA or stored through a series of enzymatic steps as retinyl esters. Once synthesized, RA is transported to the nucleus where it recognizes and binds to retinoic acid receptor–retinoid X receptor (RAR–RXR) heterodimers bound to RA response elements where they regulate transcriptional progression of various genes (wnt, hox) [2,13].

Retinoid signaling has been shown to be involved in various regenerative processes, such as zebrafish fin and heart regeneration or during *Xenopus* tadpole hindlimb regeneration [14,15]. In order to ascertain the range of species and organs where RA plays a regenerative role, it is important to extend the studies to novel species and different organs. The sea cucumber, *Holothuria glaberrima,* can regenerate its digestive system following visceral autotomy (evisceration). The holothurian digestive tract is composed primarily of a long and curly tube that is attached to the body wall by the mesenteries and that occupies most of the body cavity. The intestine is subdivided by the direction of its axis as the first descending, the ascending, and second descending intestine, or by anatomical features as small (first descending plus ascending) and large (second descending) intestine [16]. The general architecture of the holothurian intestinal system roughly resembles the vertebrate one, as it displays a luminal (digestive) epithelium, followed by the connective tissue, both muscle layers (circular and longitudinal), and a coelomic epithelium [17]. These correspond to the vertebrate mucosa, submucosa, muscle layers, and serosa.

Evisceration is a common feature to many sea cucumbers [16,18]. This process is coordinated by the nervous system and involves the rupturing of the digestive tube from the mesentery at precise zones, therefore minimizing variation in the evisceration process among members of the same species [19]. Detachment of the digestive tube from the mesenteries in our model system is induced by injecting potassium chloride (KCl) into the body cavity. KCl promotes the autonomous rupturing of the digestive tube at the esophagus and the cloacal ends followed by its subsequent expulsion with other attached organs (e.g., hemal system, respiratory tree) through the cloaca [20]. During evisceration, the gastrointestinal tissue is completely eliminated from the esophagus to the cloacae, leaving the torn edges of the mesentery healing within the body cavity.

The digestive system regeneration takes place along the complete edge of the mesentery, from the esophagus to the cloaca. Initially, multiple irregular thickenings from the torn mesentery edges can be detected between three and five days post evisceration (DPE). The 5 DPE stage is characterized by a drastic simplification of the architecture within the mesothelium that coincides with a peak in the appearance of spindle-like structures (SLSs) along the mesentery adjacent to the intestinal rudiment [21]. The appearance of SLSs is due to the condensation of the myofilaments from the muscle cells undergoing dedifferentiation and is considered a trademark of this cellular process [22]. This regenerative stage is accompanied by a surge in dividing cells along the coelomic epithelium of the intestinal rudiment that coincides with an increase in its area. The thickenings will form a solid blastema-like structure by 7 DPE, and by 14 DPE the basic layout of the intestine has been established. Finally, a fully formed alimentary tract can be identified at 21 DPE; however, by this period it still remains a portion of its original size [16]. The amazing regenerative capacity described before, places this organism as an excellent biological system to study the molecular mechanisms involved in the regenerative processes, specifically the ones related to organogenesis of the gastrointestinal system in deuterostomes.

Recently, we have identified and characterized various isoforms of the retinoic acid receptor (RAR) and the retinoid X receptor (RXR) within the sea cucumber’s transcriptome [23]. The sea cucumber receptors show several isoforms; however, they do not correspond to the alpha, beta, and gamma classification that is associated with genomic duplication in vertebrates. We have previously shown that some of the holothurian receptors were significantly over-expressed during specific stages of the intestinal regenerative response [23]. In this work we identified other sequences within the sea cucumber transcriptome that are involved in retinoid metabolism and homeostasis. Thereafter, we determined the expression patterns of these genes during the intestinal regeneration process. Finally, we suggest that RA signaling could be guiding the cellular processes that lead to the formation of the new intestinal rudiment by modulating its signal using a pan-RAR agonist (tazarotene), an RAR antagonist (LE135), and an RA synthesis inhibitor (citral). In doing this we intend to further contribute to the understanding of the role endogenous RA play in regenerative abilities, such as those observed in urodele amphibians, anuran tadpoles, and teleost fish [14,15,24]. Here we report the first analysis of the involvement of RA signaling in the regenerative prowess of the intestinal system in an echinoderm.

## 2. Materials and Methods 

### 2.1. Collection, Maintenance and Evisceration of H. glaberrima Specimens

Adult sea cucumbers (8–12 cm long) from the *Holothuria glaberrima* species were collected during low tide at rocky intertidal zones of northeastern Puerto Rico. Individuals were directly transported to the research facilities and housed in aerated seawater aquaria at room temperature (22 ± 2 °C) for at least 16 h before the beginning of the experimental procedures. Evisceration was induced by injecting 3–5 mL of 0.35 M KCl within the body cavity [20]. These individuals were allowed to regenerate for 3, 5, 7, or 14 days post evisceration (DPE) for the subsequent expression analyses or during 6 DPE for the histochemical assays.

### 2.2. Sequence Analyses: Bioinformatics and Phylogeny

Sea urchin orthologs for the short chain dehydrogenase/reductase 7 and aldehyde dehydrogenase 8 family A1 from the *Strongylocentrotus purpuratus* species (Appendix A) were probed against databases with over 5173 expressed sequence tags (EST) from normal (non-eviscerated) and regenerating (eviscerated) intestinal tissues [25], as well as over 3 million RNAseq sequences obtained from nervous and intestinal tissues [26,27]. Hits (e-value < 10^−30^) were subsequently queried against the non-redundant protein database within the NCBI webpage employing the Basic Local Alignment Search Tool (BLAST) to determine their identity and establish a proper reading frame.

Primer pairs (Table 1) were designed from consensus segments to determine the full-length cDNA sequence through multiple rounds of polymerase chain reaction (PCR) amplification and sequencing. The samples were subsequently sequenced at the Sequencing and Genotyping Facility (UPR-RP) and the resulting information was later deposited into the NIH-GenBank repository under accession numbers MN124283 (SDR7) and MN124284 (ALDH8A1). Internet-based prediction programs, SMART [28] and InterProScan [29], were used to search for conserved domains of the predicted protein sequences. Geneious 9.1 (www.geneious.com) was subsequently used to develop the multiple sequence alignments and maximum likelihood phylogenetic trees (PhyML) [30]. Information of the sequences used in the latter two analyses is included in Appendix A. The Phyre2 (Protein homology/analogy recognition engine) web-based service was used for prediction of the short-chain dehydrogenase/reductase 7 tertiary structure.

### 2.3. Gene Expression Profile

The protocols for mRNA extraction and PCR amplification have been described earlier [17]. In brief, the descending small intestine (anterior), the region connecting the ascending small intestine to the large intestine (medial), and large intestine (posterior) of the digestive tract were collected and placed in RNA later (Ambion). Total RNA was extracted using the TRI reagent (Sigma) and treated with RNase-Free DNase I (QIAGEN). Concentration and purity of the RNAs were assessed using the NanoDrop ND-1000 spectrophotometer (Thermo Scientific). First strand DNA complementary to RNA (cDNA) was synthesized from 1 µg of total RNA with the Impromp-II Reverse Transcription System (Promega) and oligo (dT)_23_ primer. PCR primers were designed using PrimerQuest and OligoAnalizer tools from the Integrated DNA Technology webpage (www.idtdna.com). Semiquantitative expression was assessed from the optical density values (Quantity One 4.6.6—BioRad) of the images taken with the Molecular Imager ChemiDoc XRS+ (BioRad) from the samples bands in the electrophoresis gels normalized against the optical density values of NADH dehydrogenase subunit 5 [25,31]. PCR products were collected at a cycle depicting the beginning of their logarithmic phase. Normal individuals were kept in the laboratory aquaria and at least one was sacrificed with the regenerating individuals at each of the sampled stages: 3, 5, 7, and 14 DPE.

### 2.4. Drug Treatments

Organisms were eviscerated and then allowed to recover for an hour before being transferred into aerated aquaria with 800–1000 mL of seawater in groups of four. All drugs were first diluted in DMSO and subsequently added directly to the seawater in each aquarium so as to reach the target concentration. Studied groups includes: (1) the retinoic acid receptor β/γ selective agonist, tazarotene {6-[2-(3,4-Dihydro-4,4-dimethyl-2*H*-1-benzothiopyran-6-yl) ethynyl]-3-pyridinecarboxylic acid ethyl ester) at 33 μM, (2) the retinoic acid receptor β antagonist, LE135 {4-(7,8,9,10-tetrahydro-5,7,7,10,10-pentamethyl-5*H*-benz[e]naphtha [2,3-b][1,4]diazepin-13-yl)-benzoic acid} at 10 μM, (3) a competitive inhibitor of aldehyde dehydrogenase, citral (3,7-dimethyl-2,6-octadienal) at 25 μM, and (4) a control group containing the vehicle employed for drug dilution (added the equivalent to the highest volume of vehicle used for the dilutions, v/v), DMSO (dimethyl sulfoxide). All drugs were obtained from Sigma, except Tazarotene, which was acquired from Tocris (Bristol, UK). Each aquarium was individually covered with aluminum foil to minimize light exposure since citral is light sensitive. Seawater was replaced daily with fresh seawater and the corresponding drugs during the following 5 days. Animals were injected with BrdU (5-bromo-2′-deoxyuridine—Sigma—50 mg/kg body weight) in the coelomic cavity at the 6 DPE stage and placed back in their aquaria for 5 h; animals were then sacrificed for further processing and analysis (CITI Program—Working with Animals in Biomedical Research Record Id 32767248).

### 2.5. Histology: Tissue Fixing, Sectioning, and Labeling

Regenerating and normal organisms were anesthetized, by placing them in 0.5% chlorobutanol (1,1,1-trichloro-2-methyl-2-propanol hydrate—Sigma) diluted in seawater, for 20–30 min, sacrificed, and dissected across the dorsoventral ambulacra. The intestinal rudiment attached to the body wall of the animal was then fixed overnight in 4% paraformaldehyde diluted in 0.1 M phosphate-buffered saline (PBS) at 4 °C. The tissues were rinsed three times with 0.1 M PBS after a ~24 h fixing period and stored in 30% sucrose/0.1 PBS at 4 °C until sectioned. Tissue samples were embedded in OCT medium (Tissue Tek OCT—Sakura Finetek, Torrance, CA, USA), frozen, and cross sectioned in a Leica CM1850 cryostat at 20 μm. Immunohistochemical labeling of dividing cells with BrdU [16] or muscle labeling with fluorescently-labeled phalloidin [22,32] have been previously described. To determine the level of cell division, slides were placed in a humid chamber, washed with PBS-Triton 100× (1%) for 15 min, rinsed with 0.1 M PBS, and treated with HCl 0.05 M for 1 h. Thereafter the slides were exposed to the primary antibody (mouse anti-BrdU—GE Healthcare—1:5) during a ~24 h incubation period, rinsed thrice with 0.1 PBS, and incubated for an hour with goat anti-mouse labeled with Cy3 (GAM-Cy3—Jackson ImmunoResearch Laboratories, West Grove, PA, USA—1:1000). Finally, the slides were washed with PBS three times, and mounted in buffered glycerol containing DAPI. Muscle labeling was done by incubating tissue sections with fluorescent-labeled phalloidin-TRITC (phalloidin-tetramethylrhodamine B isothiocyanate—Sigma—1:2500). Phalloidin tightly binds actin molecules present in muscle. Incubation was done for an hour, then sections were washed with 0.1 PBS three times, and mounted. Images were observed and documented using a Nikon Eclipse Ni microscope equipped with a Nikon DS Qi2 camera (Nikon Instruments, Melville, NY, USA). Cell division was determined as the ratio of cells labeled with BrdU along the blastemal coelomic epithelium divided by the total amount of cells labeled with DAPI within the same region. Cell dedifferentiation was determined as the ratio of spindle-like structures (SLS’s), a hallmark for muscle dedifferentiation, present within the coelomic epithelium along the mesenteries near the blastema divided by the total number of DAPI-labeled cells along the same region. At least three technical replicates and three biological replicates were performed per group for the histological analyses.

### 2.6. Statistical Analysis

Statistical significance of the resulting data was evaluated through ordinary one-way ANOVA, which included a Tukey’s multiple comparisons test for a single pool variance. All values were reported as the mean ± standard error while a *p* < 0.05 was considered to indicate statistical significance difference between groups. The statistical analyses were performed in GraphPad Prism 6.01 (GraphPad Software, San Diego, CA, USA). For the mRNA expression profiles, three biological replicates were analyzed for each regenerating stage and four for the normal gut.

## 3. Results

### 3.1. Sequence Characterization of Two Alcohol Dehydrogenases in the Sea Cucumber

Entries with significant similarity (E value < 10^−30^) to deuterostome orthologs for short chain dehydrogenase/reductase 7 (SDR7) or aldehyde dehydrogenase 8 family A1 (ALDH8A1) were recognized within our transcriptome database (compiled from intestinal and nervous tissues of the sea cucumber *Holothuria glaberrima*). The sequence information was then used to design primer pairs (Table 1) to amplify and sequence these transcripts from cDNA synthesized from normal and regenerating intestinal tissue. The obtained sequences were analyzed and shown to correspond to the holothurian orthologs of these two dehydrogenases. 

Our analysis shows that the sea cucumber’s SDR7 sequence has an ORF of 984 base pairs (bp) that encodes for a predicted protein of 327 aa (Figure 1). We also identified a small segment of the UTRs at both the 5′ (21 bp) and 3′ (76 bp) ends (Figure 1). The encoded protein contains the family characteristic cofactor binding site [33] along residues 74–81 (TGxxxGxG) and the catalytic tetrad along residues 181, 209, 222, and 226 (N-S-Y-K), both within the NAD(P)(H)-dependent oxidoreductase site. Further sequence motifs previously described in other species [33] and found within the sea cucumber’s SDR ortholog include a singular Asp at aa 130 associated to the stabilization of the adenine ring pocket and weak binding of the coenzyme; an NNAG sequence along aa 156–159 important in the maintenance of the central β sheet; a PG motif among aa 253–254; and a Thr at aa 257 associated with cofactor interaction (Figure 2). The hypothetic three-dimensional structure of the sea cucumber’s SDR7 sequence was predicted using the web-based program Phyre2, which depicts the alpha/beta-folding and the central beta-sheet pattern (Rossmann-fold) characteristic of the SDR family (Figure 3). A multiple sequence alignment comparing our sequence to other SDR7s representatives from major metazoan phylogenies demonstrated that our sequence displays the highest overall identity (46%) and similarity (70%) to the SDR7 sequence of another echinoderm, the sea urchin *Strongylocentrotus purpuratus* (Table 2).

The nucleotide sequence of ALDH8A1 is 1212 bp long, which translates into a 403 aa protein (Figure 4). Parts of the 5′UTR (131 bp) and 3′UTR (172 bp) sequences were also obtained (Figure 4). The *H. glaberrima* ALDH8A1 ortholog displays a number of highly conserved residues along the NAD binding site that distinguish the members of the ALDH superfamily [34]. These motifs include residues essential for catalysis, such as the Asn at aa 171, the Glu at aa 269, the Gly at aa 300, and the Cys at aa 303, or for cofactor binding, such as the glycines of the Rossmann fold (GxxxxG) present at aa 247 and 252. Additional cofactor binding residues identified within our transcript include a Lys at aa 194, a Glu at aa 408, and a Phe at aa 410 (Figure 5). A multiple sequence alignment comparing the sea cucumber’s sequence against representatives from major metazoan phylogenies showed our sequence displays an identity of 55% and a similarity of 69% to the starfish *Acanthaster planci*, another echinoderm (Table 2).

### 3.2. Phylogenetic Analysis of the Sea Cucumber’s SDR7 and ALDH8A1 Display Ambulacral Ancestry

The *H. glaberrima* SDR7 sequence was probed against orthologs from different metazoans. This analysis encompassed sequences from other SDR families to serve as outliers. In the resulting maximum likelihood phylogenetic tree SDR7 sequences clustered together with a bootstrap value of 88% out of 1000 replicates. Further examination shows that *H. glaberrima* SDR7 sequence grouped alongside acorn worm (hemichordata) and sea urchin (echinoids); both members of the Ambulacria (a basal group of deuterostomes that includes echinoderms and hemichordates and are closely related to chordates). Moreover, each phylogenetic group included in the analysis (chordates, lophotrochozoan, and ecdysozoans) formed explicit clusters within the tree. Meanwhile, the short branches amongst the multiple SDR7s analyzed suggest small evolutionary changes (defined by substitutions per site) in this group when compared to the other alcohol dehydrogenases families. The other SDRs families included in the analysis grouped together at the bottom of the phylogram (Figure 6).

A similar analysis performed for the deduced sequence of ALDH8A1 isolated from the sea cucumber resulted in a similar outcome. In this maximum likelihood phylogram the ALDH8A1 sequences clustered together at the bottom of the tree with a bootstrap value of 80%; the sea cucumber sequence clustered within the holothurian clade, alongside the sea urchin and the starfish. This clade was flanked by the chordates towards the bottom branches of the tree and by protostomes (lophotrochozoa and ecdysozoa) and cnidarian towards the top branches. Upon further examination every other ALDH8A1 clustered within the branch characteristic of its phylogenetic group: chordates, lophotrochozoans, ecdysozoa, and cnidarian. Other aldehyde dehydrogenases from the sea urchin and retinaldehyde dehydrogenases (RALDHs) from the mouse were used as outliers, both clustering at the top of the tree with a bootstrap value of 46% (Figure 7).

### 3.3. Quantitative Assessment of Transcript Abundance Showed Regulation of SDR7 during Intestinal Regeneration

We explored, using semi-quantitative reverse transcription PCR analysis, if the relative expression of the SDR7 and ALDH8A1 transcripts were associated to specific regions of the digestive tract (anterior, medial, or posterior). The results showed a small but statistically significant increase in expression of the SDR7 transcript along the medial and posterior intestinal segments (Figure 8). Meanwhile, no significant difference was detected for the ALDH8A1 transcript (Figure 9). Statistically significant expression of the SDR7 transcript was observed at the 7 and/or 14 DPE stages when compared to its expression at the 3 DPE one (Figure 8).

### 3.4. RA Signaling Modulation Affects Overall Rudiment Size during Regenerative Organogenesis of the Intestine

To explore if the RA signaling pathway is involved in the cellular processes linked to intestinal regeneration, we eviscerated animals and allowed them to regenerate in the presence of pharmacological drugs known to modulate the RA signaling pathway. The drugs used for this analysis included a RAR agonist (tazarotene), a RAR antagonist (LE135), and a noncompetitive inhibitor of ALDH (citral). After being exposed to the different drugs or dimethyl sulfoxide (vehicle) for the first 6 days of regeneration, animals were injected with BrdU five hours before being sacrificed. The regenerating organs were then fixed and processed for histological analysis. Finally, the effects of the drug were assessed by comparing the size of the regenerating rudiment among the various groups. This provides a quantifiable evaluation of whether regeneration is being inhibited or accelerated. No effect was observed when the RAR agonist (tazarotene) was compared to the control. However, both the ALDH inhibitor (citral) and the RAR antagonist (LE135) significantly reduced the size of the regenerating rudiment to one third and one sixth of the control size, respectively (Figure 10).

### 3.5. RA Signaling Modulation Affects Cellular Processes Associated with the Regenerative Organogenesis of the Intestine

Two possible cellular mechanisms that are known to play important roles in *H. glaberrima* intestinal regeneration are cell dedifferentiation and cell proliferation [21]. To determine whether cell division was affected by the drugs modulating the RA signaling pathway, we used BrdU to quantify the percentage of dividing cells in the intestinal rudiment of control and experimental organisms. Similarly, to determine whether cell dedifferentiation was affected, we measured the relative numbers of SLS (a marker of muscle cell dedifferentiation) [22]. Our results showed no significant effect of tazarotene (~20% cell division) when compared to the control (~19%). However, both RA signaling pathway inhibitors almost halved the percentage of dividing cells (LE135 ~8% or citral ~10%—Figure 11). Similar results were obtained when comparing the dedifferentiation process, where the percentage of SLSs present in the mesenteries near the rudiment area of the groups treated with tazarotene (~34%) showed no significant differences to the control (DMSO—35%). However, the level of dedifferentiation is less than half in LE135 (~14%) and citral (~16%) treated animals when compared to the control (Figure 12).

## 4. Discussion

### 4.1. Holothurian Enzyme Orthologs

Here we have identified the sequences of two enzymes involved in the metabolism of RA from a transcriptomics database of the sea cucumber, *Holothuria glaberrima*. Evidence shows that the core structure of the SDR7 peptide found in *H. glaberrima* (327 aa) is within the typical (250–350 aa) length of members from the SDR superfamily [33]. SDRs constitute one of the largest (>47,000 members) and functionally heterogeneous protein superfamilies known [35]. Phylogenetic comparisons of this group exhibit early divergence as the majority of its members have low pairwise sequence identity; still they share common sequence motifs and are virtually present in every genome explored [36]. The sea cucumber SDR7 retains the characteristic NADP(H)-dependent oxidoreductase site harboring the glycine rich cofactor binding site and the catalytic tetrad. Furthermore, the general architecture of the holothurian SDR includes other characteristic sequences, such as cofactor interaction sites or motifs important in the maintenance of its tertiary structure [33]. Three-dimensional structure prediction of our sequence depicts the Rossmann-fold, a distinctive characteristic within the members of this group [32,33]. Furthermore, the holothurian SDR sequence groups within the ambulacral clade of the SDR7 branch in the maximum likelihood phylogenetic analysis. Interestingly, SDR7 is one of four families that standout with retinol and steroid dehydrogenase activity [37].

Evidence confirming the identity of the sea cucumber’s ALDH8A1 shows this sequence conserves the NAD(H) oxidoreductase region harboring the catalytic and cofactor binding residues that characterize the ALDH superfamily [34]. Members of this clade, along with the ALDH1A1-3 family, participate in the biosynthesis of RA through the irreversible oxidation of all-trans or 9-cis retinal [34,38]. Further structural evidence shows the presence of the GxxxxG motif which depicts the Rossmann-fold involved in cofactor binding while the phylogenetic analyses cluster our sequence along the echinoderm clade within the ALDH8A1 specific branch. The bioinformatics analyses strongly suggest that the characterized enzymes are the sea cucumber homologs for SDR7 and ALDH8A1 and that these enzymes are part of a *bona-fide* retinoid mediated signaling pathway.

### 4.2. Differential Expression of Genes Linked to RA Signaling during the Regenerative Response

It is widely known that RA is an essential regulator of vertebrate development [13,39,40], while several lines of evidence also support an active role for this molecule in various regenerative events. For example, dorsal fin amputation in the zebrafish induces RA signaling through the overexpression of the RDH10B, ALDH1A2, and RARα transcripts in the stump tissue [41]. Additional evidence for the involvement of this signaling pathway in regenerative events comes from experiments in newts after tail resection where the expression of a RAR variant (RARβ2) was evident following tail amputation [24]. Here we showed that the sea cucumber SDR7 ortholog rapidly bounces up to control values seven days following evisceration. Interestingly, previous work from our group showed that one of the retinoic acid receptors (RAR-Long) also increases significantly from three to seven days following evisceration [23]. These increases in both RA pathway members coincides with the increase in cell proliferation that takes place in the regenerating intestinal rudiment [20,21]. Thus, the differential expression of the SDR7 and RAR-L transcripts could suggest that regulation of the machinery linked to RA synthesis might be employed as a method for controlling the levels of RA available during the regenerative events of the gut in this basal deuterostome in a way similar to what takes place during the regenerative processes in zebrafish, *Xenopus*, or newt. Furthermore, the results from ALDH8A1 suggest that this particular enzyme might not be involved in the regenerative response. This is in line with other research that showed no de novo expression of ALDH1A1-3 during hindlimb bud regeneration in the *Xenopus* tadpole [14].

Nonetheless, it is important to remember that the control of RA production can take place at many different levels. For one, we are measuring the levels of mRNA using a semi-quantitative method and that the significant differences of the retinoid transcripts do not necessarily mean correlative changes in protein levels. Moreover, the effect of a metabolic pathway not only hinges on the protein concentration but also on the levels of enzyme activity that can be modulated without any changes in the concentration of mRNAs or protein. Furthermore, we have identified only two sequences associated to RA anabolism in the sea cucumber’s transcriptome. Many other transcripts are involved in mediating retinol oxidoreduction. Retinol oxidation can be mediated by the action of specific members from a diverse group of medium-chain alcohol dehydrogenases (MDR-Adh) and short-chain retinol dehydrogenases (SDR-Rdh) [42,43,44,45]. Sequences such as SDR9, SDR11, SDR16, ADH1, ADH2, ADH4, ADH7, RDH4, and RDH10 stand out among those shown to mediate retinol oxidoreduction [2,4,37] but have not been identified so far within our database. Nevertheless, some of these have been identified within the genome of the sea urchin, *Strongylocentrotus purpuratus* (the closest phylogenetic representative with a fully sequenced and well annotated genome). Therefore, some of these are most likely present within the sea cucumber’s genome and could be influencing retinoid synthesis during regenerative processes. Moreover, the mRNA of other enzymes affecting RA availability such as the cellular RA binding protein 2 (CRABP2) and cytochrome P450 family 26b (Cyp26b) have been found to be differentially expressed at specific stages of *Xenopus* tadpole hindlimb regeneration [14], yet the former has not been identified within our database so far while the latter is currently being characterized.

### 4.3. Functional Depiction of a RA Signaling Pathway during Intestinal Regeneration

Pharmacological tools have been commonly used to modulate the RA signaling pathway in animal models as a mean to gain insight into specific cellular processes. For example, the role of endogenous RA during limb regeneration in the axolotl has been probed using citral (100 μM) which caused a slower rate of forelimb regeneration while severely impairing limb patterning. Organisms treated with smaller concentrations of the drug (10 or 30 μM) were indistinguishable from the controls [46]. Further evidence links RA signaling to the regenerative response of the intestinal system in a budding ascidian model system. In this model, citral inhibited the formation of the posterior half of the gut (esophagus, stomach, intestine) following amputation [47]. Here we showed that, as in the axolotl, the sea cucumber’s regenerating intestinal rudiment is smaller in size when treated with citral (25 μM), nevertheless it did not completely suppress regeneration, as in the budding ascidian, nor resulted in any observable patterning defects as in the amphibian. Inhibition of regeneration has also been documented by treatment with a RARβ antagonist (LE135). Use of this inhibitor at 1 μM caused a decrease in the overall tail length during the regenerative events of a urodele amphibian [24]. Furthermore, application of LE135 (250 nM) truncated limb regeneration in the axolotl without affecting blastema formation. Similar results have been shown using other RA receptor antagonists. For example, treatment with a different RAR antagonist (LE540) negatively affected the size of the axolotl blastema [48]. Our results in the holothurian model system showed that visceral regenerative organogenesis is also impaired when RA signaling is hindered or inhibited, therefore strengthening the role of RA in regenerative events.

On the other hand, our results differed when trying to enhance the RA signaling system. In the axolotl forelimb following amputation, application of different concentrations of an RA precursor (retinyl palmitate) produced extra elements; a phenomenon that was also witnessed during hindlimb regeneration [49]. Treatment with the same RA precursor also promoted the formation of hindlimb segments not lost during amputation in *Rana temporaria*, and a higher dose caused the duplication of the entire limb [49]. More recent work has shown that over-activation of the RARβ variant with a pharmacological agonist (AC55649 50 mM) induced bilateral hindlimb duplication during the regenerative events of *Xenopus laevis* [50]. Moreover, treatment with tazarotene (1–10 μM) promoted endothelial tube remodeling and branching in an organotypic angiogenesis assay using human umbilical vein endothelial cells [51]. Al Haj Zen and colleagues also showed that treatment with tazarotene (10 mg/kg) promoted wound healing and neovascularization in an ear punch mouse model [51].

These experiments contrast with our finding that application of tazarotene (33 μM) had no effect on the overall size of the intestinal rudiment or on cell division or dedifferentiation. Based on this information it seems that overactivation of RAR does not promote intestinal regeneration in our model system. Nonetheless, alternative explanations for the lack of response could be first that the drug does not activate echinoderm (or holothurian) receptors, second that the cofactor complexes modifying local chromatin structure or engaging the basal transcription machinery are not present nor being recruited, or third that a higher dose might be needed for the proper activation of this signaling pathway. Yet the concentration of tazarotene employed in our research is similar or higher than those employed in the studies mentioned above.

Cellular dedifferentiation is a common response in the regenerative process of many animal species [52]. Similarly, regeneration of the sea cucumber intestine involves massive reorganization of the mesenteries that relays on the plasticity of the mesothelial myoepithelial cells undergoing dedifferentiation [16]. These cells condense and expel their myofilaments into compact SLSs, which allow them to migrate, proliferate, and redifferentiate, helping in the formation of the new intestinal rudiment [21,22,53]. Here we showed that interfering with RA signaling through the application of citral or LE135 significantly decreases cell dedifferentiation along the mesenteries while also diminishing cell proliferation in the mesothelium of the blastema during intestinal regeneration. Altering these and possibly other cellular processes through the application of these drugs had a negative effect in the overall size of the regenerating intestinal rudiment. RA has been shown to be a key signaling pathway influencing cell cycle entry of formerly quiescent cells in a zebrafish fin regeneration model since the application of exogenous RA promoted proliferation in the fin stump by 30% when compared to DMSO. Moreover, inhibition of the RA signal through the activation of a Cyp26 or a dominant negative RAR transgene construct respectively diminished cell proliferation by 89% or 83% when compared to the wild-type in the same model system [41]. RA treatments have also been linked with cell proliferation in mice since the application of a single does (25 μg/g) caused hepatomegaly while inducing transcription of cell cycle genes in normal liver following a sham operation. Furthermore, regenerating liver in RA-treated mice displayed early induction of RA signaling sequences (e.g., RARβ, ALDH1A2, CRABP1) and accelerated liver regeneration following partial hepatectomy. This response coincided with a significant increase in the number of proliferating cells along RA treated livers [54]. In summary, two of the principal underlying mechanisms guiding the regenerative response in various regenerative models, cell dedifferentiation and proliferation, seems to be mediated through the signaling of RA in our model system. This suggests that the RA signaling pathway could be an important component guiding the cellular processes linked to the regenerative response of the intestine in this model system.

## 5. Conclusions

Here we have presented the first research work associating the RA signal to the regenerative process of the sea cucumber intestine. We have shown that this signal affects the overall size of the regenerating intestinal rudiment by modulating cellular dedifferentiation and division—two processes linked to the regenerative response in other model systems. Still, we have only characterized the expression of two sequences associated to retinoid oxidoreduction and five sequences associated with its signal. Thus, we know that other sequences influencing the RA signal are present in the sea cucumber’s genome. Therefore, only when we have incorporated a comprehensive analysis of these components, will we fully understand the repercussions this signal has on the regenerative response of the sea cucumber’s intestinal tissue. 

## Figures and Tables

**Figure 1 biomolecules-09-00873-f001:**
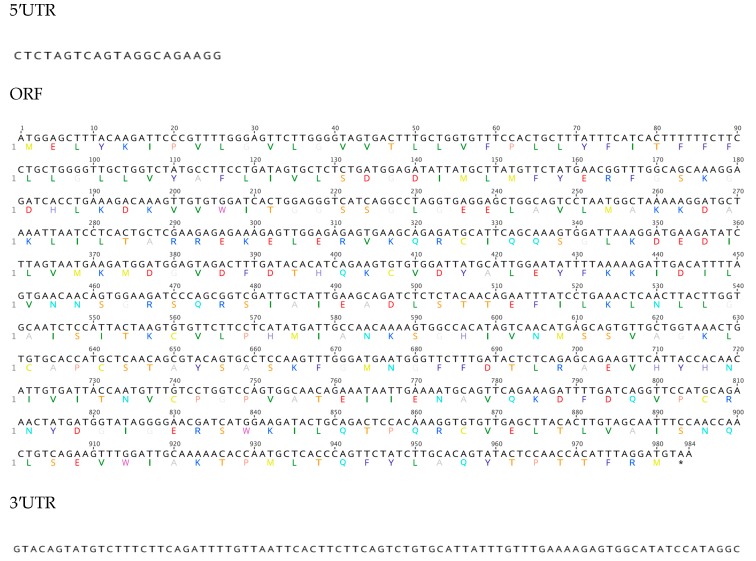
Nucleotide and amino acid sequences of short-chain dehydrogenase reductase found in the sea cucumber, *H. glaberrima*.

**Figure 2 biomolecules-09-00873-f002:**
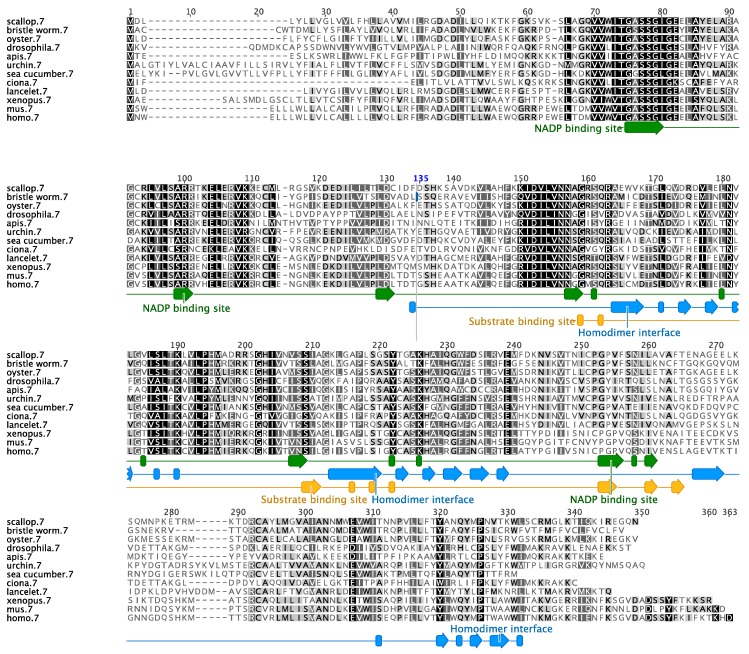
Multiple sequence alignment of the sea cucumber short-chain dehydrogenase reductase 7 amino acid sequence and those of selected protostomes, deuterostomes, chordates, and vertebrates. Conserved residues are shaded in black when they are 100% similar, dark grey if they are 99–80% similar, light grey when 79–60% similar, or white when less than 60% similar. The characteristic functional sites and interfaces of the protein are depicted herein.

**Figure 3 biomolecules-09-00873-f003:**
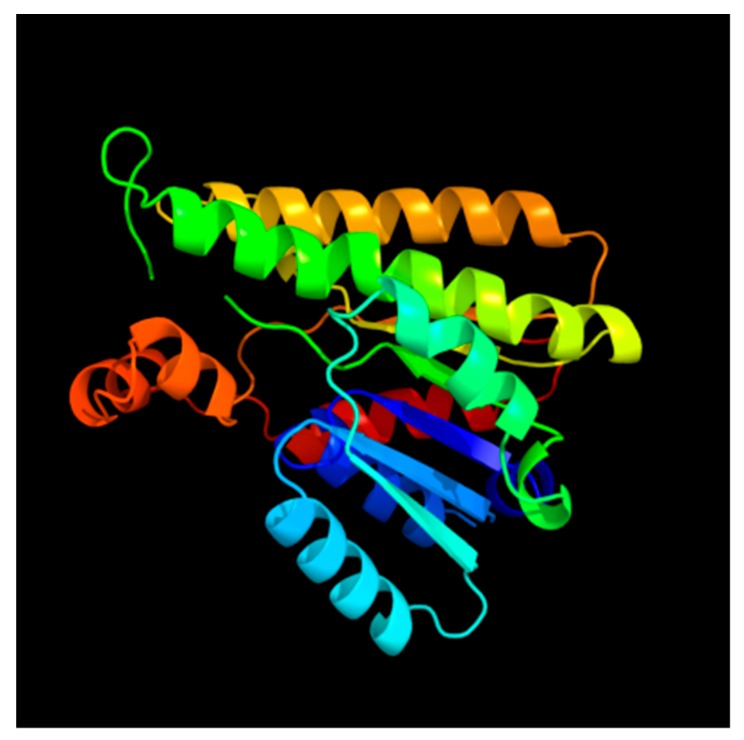
Predicted 3-D folding pattern of the sea cucumber SDR7 sequence using the Phyre2 program.

**Figure 4 biomolecules-09-00873-f004:**
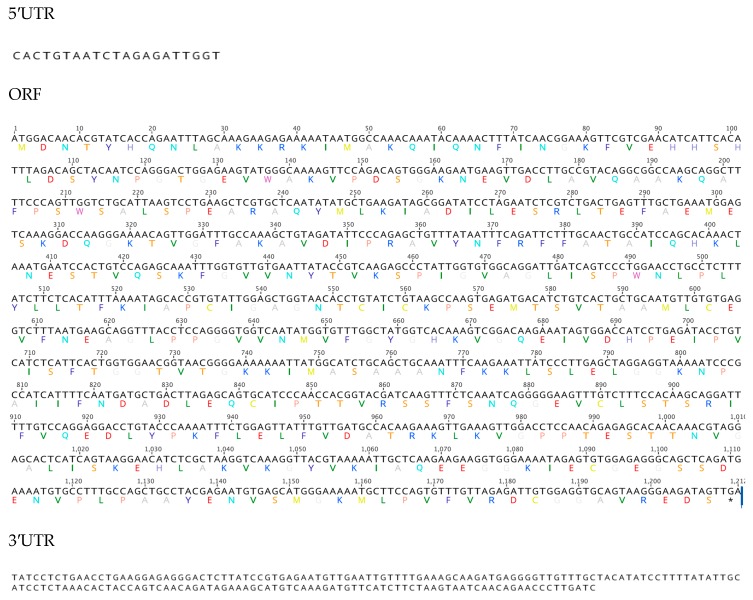
Nucleotide and amino acid organization of aldehyde dehydrogenase 8A1 found in the sea cucumber, *H. glaberrima*.

**Figure 5 biomolecules-09-00873-f005:**
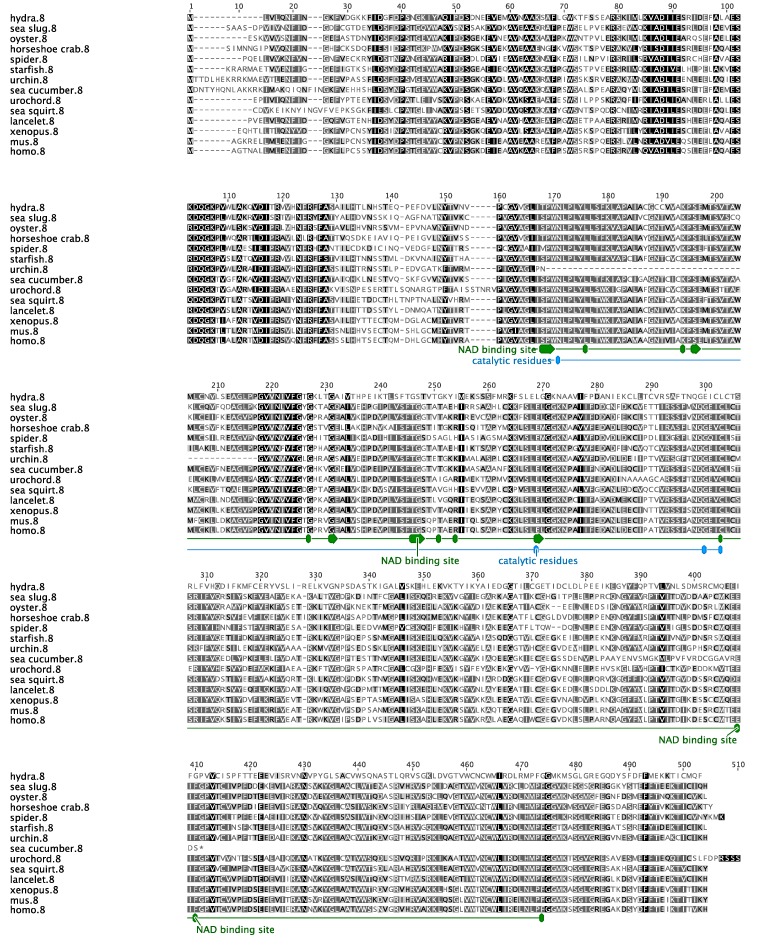
Multiple sequence alignment of the sea cucumber aldehyde dehydrogenase family 8A1 amino acid sequence and those of selected metazoan species. Conserved residues are shaded in black when they are 100% similar, dark grey if they are 99–80% similar, light grey when 79–60% similar, or white when less than 60% similar. The characteristic functional domains and sites of the protein are depicted herein.

**Figure 6 biomolecules-09-00873-f006:**
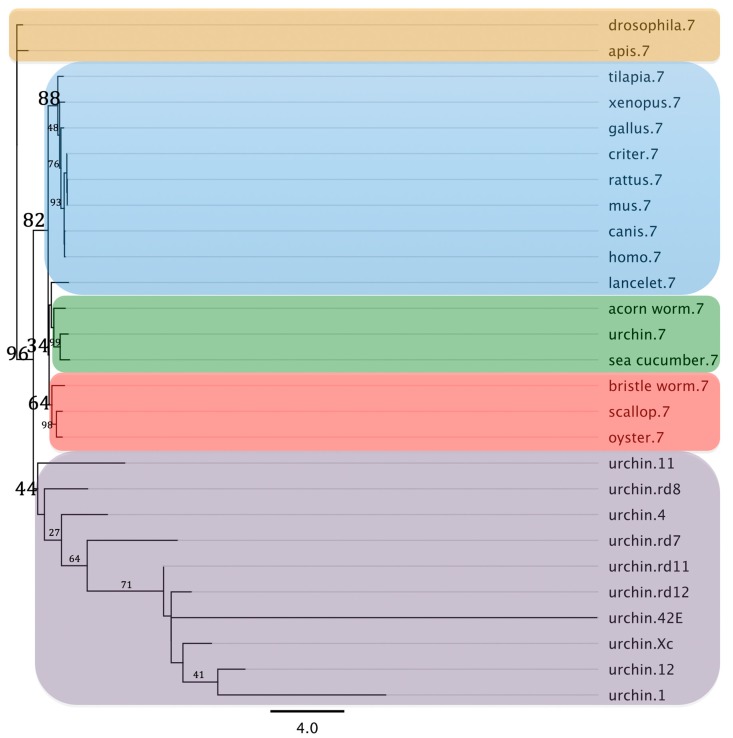
Maximum likelihood phylogram based on the amino acid sequences of the short chain dehydrogenase reductase 7 found in the sea cucumber, *Holothuria glaberrima*, compared against homologs from multiple species, other SDRs families, and retinol dehydrogenases (RD) from the sea urchin. Every SDR7 localize to the top of the phylogram and is identified by a “.7” suffix while every other SDR localize to the bottom of it and are identified by the specific family number or letter suffix. The retinol dehydrogenases also localize to the bottom of the phylogram and are identified by a “.rd#” suffix. Every sequence used for this analysis is referenced in Appendix A. Major taxonomic groups are defined by color boxes (chordates = blue; non-chordate deuterostomes = green; Lophotrochozoa = red; Ecdysozoa = beige; and other SDRs or RDs = grey) and by the bootstrap values depicted in a larger font. The numbers at the nodes correspond to the bootstrap proportion expressed as the percentage of 1000 replicates while the length of the branches is proportional to the amount of inferred evolutionary change as defined by the bar in substitutions per site.

**Figure 7 biomolecules-09-00873-f007:**
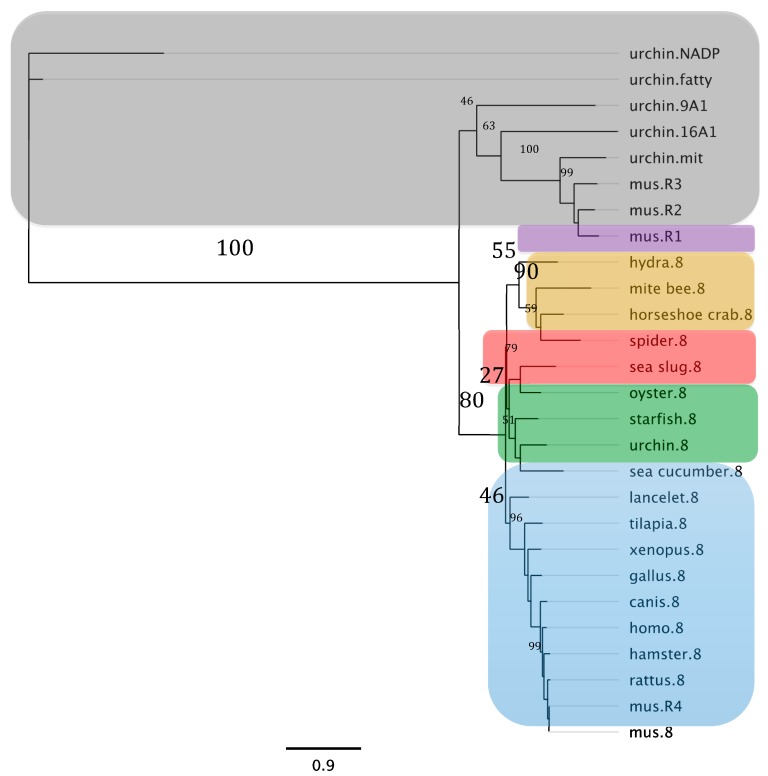
Maximum likelihood phylogram based on the amino acid sequences of the aldehyde dehydrogenase 8A1 (ALDH8A1) found in the sea cucumber, *Holothuria glaberrima*, compared against its homologs in multiple species, other ALDH8A1 families from the sea urchin, and retinaldehyde dehydrogenases (RALDH) from the mouse. Every ALDH8A1 localize to the bottom of the phylogram and is identified by a “.8” suffix while every other ALDH localize to the top of it and are identified by the specific family number or letter suffix. The RALDHs also localize to the top of the phylogram and are identified by a “.R#” suffix. Every sequence used for this analysis is referenced in Appendix A. Major taxonomic groups are defined by color boxes (chordates = blue; non-chordate deuterostomes = green; Lophotrochozoa = red; Ecdysozoa = beige, Cnidaria = purple; and other ALDHs or RALDHs = grey) and by the bootstrap values depicted in a larger font. The numbers at the nodes correspond to the bootstrap proportion expressed as the percentage of 1000 replicates while the length of the branches is proportional to the amount of inferred evolutionary change as defined by the bar in substitutions per site.

**Figure 8 biomolecules-09-00873-f008:**
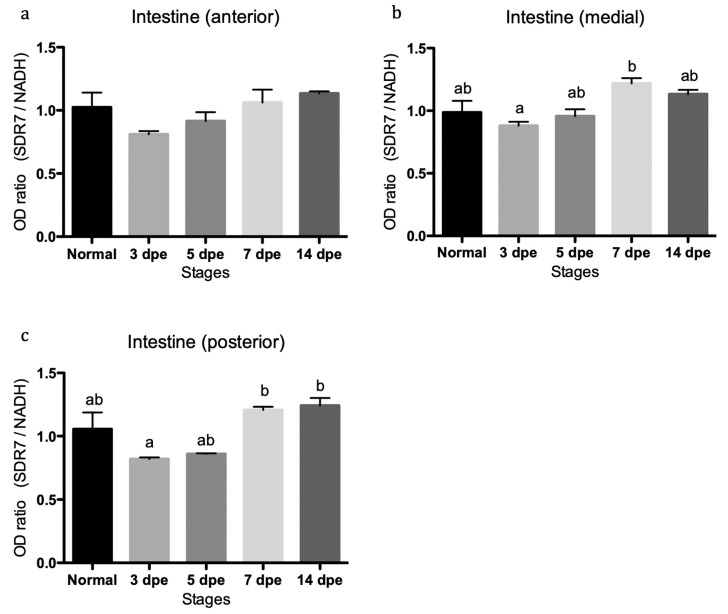
mRNA expression profiles of short chain dehydrogenase reductase 7 includes different regenerative stages (3, 5, 7, and 14 days post evisceration) and normal intestinal tissue. Semi-quantitative RT-PCR of the SDR7 transcript along the (**a**) anterior, (**b**) medial, and (**c**) posterior segments of the intestine. Three biological replicates were analyzed at each regenerating interval and four in the normal gut. Ordinary one-way ANOVA showed differences among means statistically significant in graphs b (*p* = 0.02), F = 3.5, df(treatment) = 4, df(residual) = 10 and c (*p* = 0.01), F = 5.3, df(treatment) = 4, df(residual) = 11. A multiple comparison test (Tukey) showed significant difference (*p* < 0.5) between groups bearing different letters in these two graphs. Those groups bearing two letters are not statistically different to those with any of the letters. *n* = biological replicates.

**Figure 9 biomolecules-09-00873-f009:**
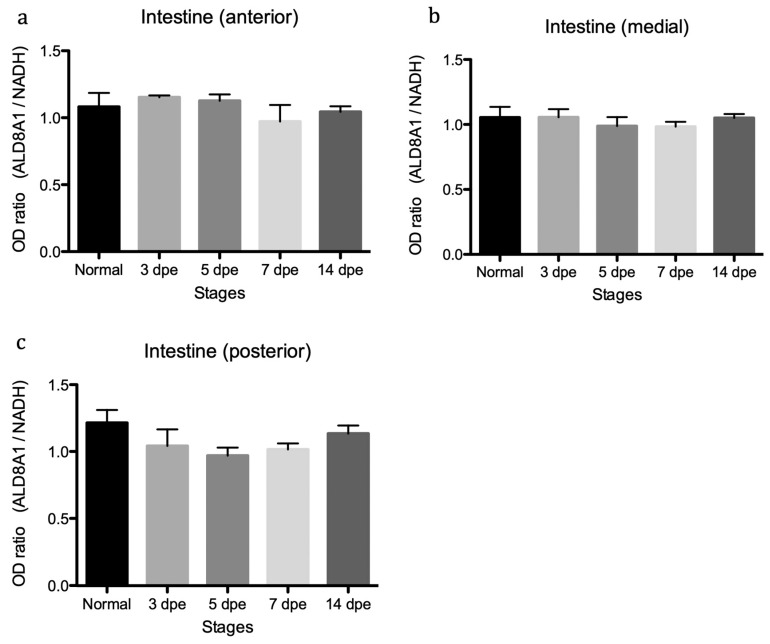
mRNA expression profiles of aldehyde dehydrogenase family 8A1 includes different regenerative stages (3, 5, 7, and 14 days post evisceration) and normal intestinal tissue. Semi-quantitative RT-PCR of the ALDH8A1 transcript along the (**a**) anterior, (**b**) medial, and (**c**) posterior segments of the intestine. Three biological replicates were analyzed at each regenerating interval and four in the normal gut. Ordinary one-way ANOVA showed no significant difference among the groups in each graph (*p* = 0.2), F = 1.5, df (treatment) = 4, df(residual) = 10. *n* = biological replicates.

**Figure 10 biomolecules-09-00873-f010:**
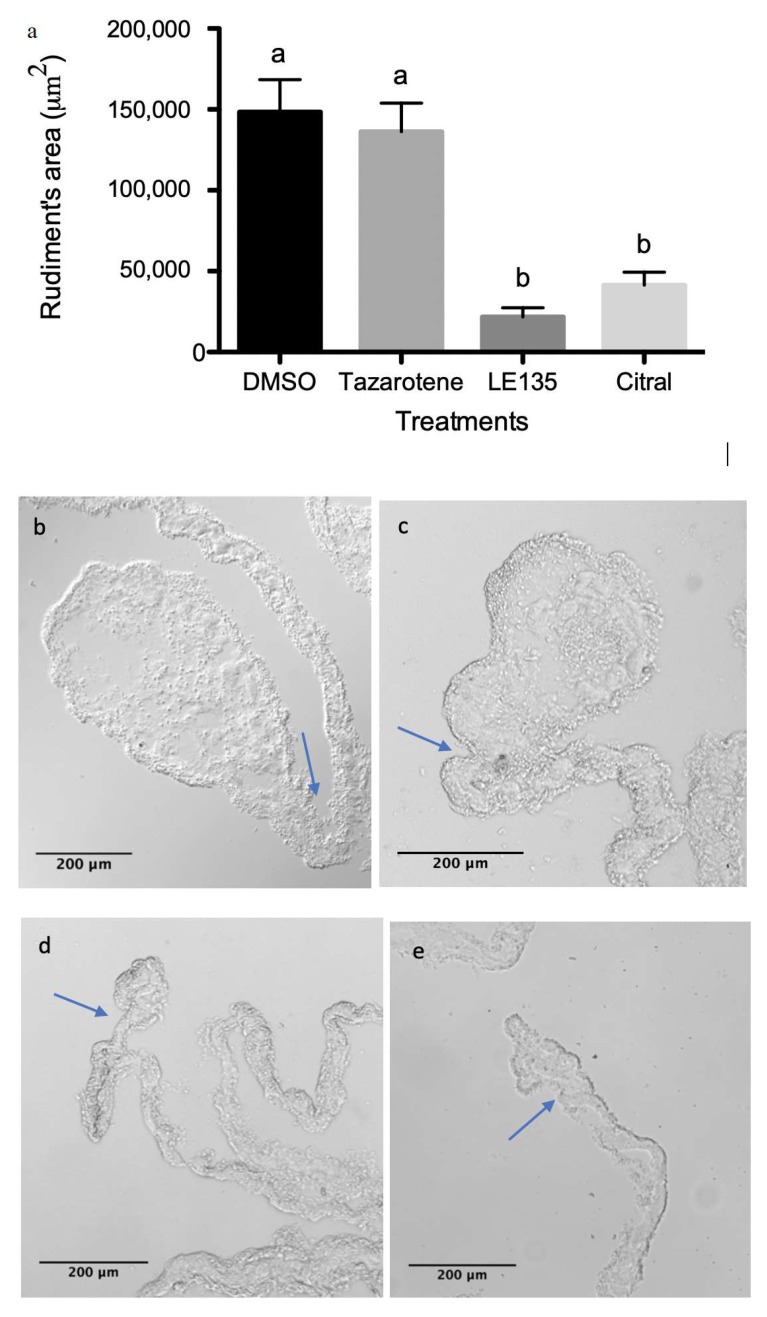
Quantification of the intestinal rudiment area of regenerating individuals. (**a**) The area of the tissue section was measured on individuals treated with (**b**) DMSO—vehicle, (**c**) tazarotene—RAR agonist, (**d**) citral—RALDH inhibitor, or (**e**) LE135—RAR antagonist following six days post evisceration. Five technical replicates (different slide sections of the same animal) and at least four biological replicates were performed per group: DMSO *n* = 10, tazarotene *n* = 4, citral *n* = 10, LE135 *n* = 4. Ordinary one-way ANOVA showed differences among means statistically significant with a *p* value < 0.0001, F = 15.87, df(treatment) = 3, df(residual) = 22. A multiple comparison test (Tukey) in these showed significant difference when comparing DMSO against citral or LE135 (*p* < 0.001) or tazarotene against citral or LE135 (*p* < 0.01). Statistically significant difference between groups is represented by different letters atop of the bars. Arrows signal the boundary between the mesentery and the new intestinal rudiment. *n* = biological replicates.

**Figure 11 biomolecules-09-00873-f011:**
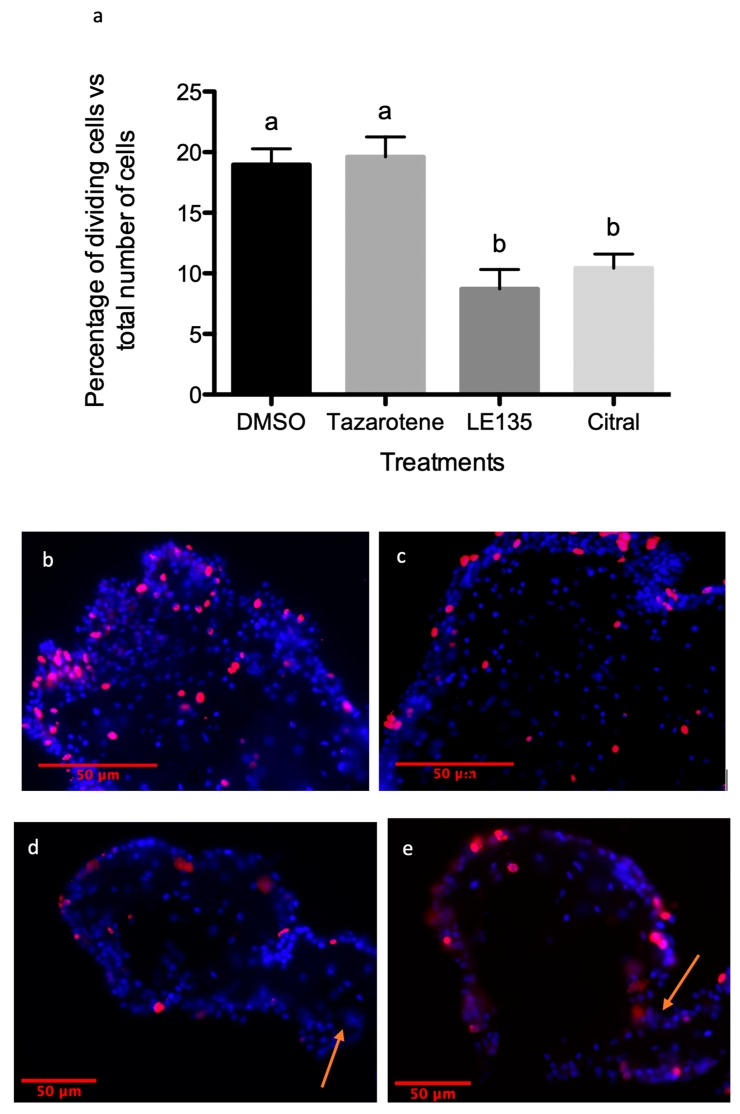
Cell proliferation along the regenerating blastema of the intestinal rudiment. (**a**) The percentage of dividing cells along the rudiment’s mesothelium was established from tissue sections labeled with an antibody against BrdU (red) on individuals treated with (**b**) DMSO—vehicle, (**c**) tazarotene – RAR agonist, (**d**) citral—RALDH inhibitor, or (**e**) LE135—RAR antagonist following 6 days post evisceration. Between three and five technical replicates and at least three biological replicates were performed per group: DMSO *n* = 8, tazarotene *n* = 3, citral *n* = 10, LE135 *n* = 4. Ordinary one-way ANOVA showed differences among means statistically significant with a *p* value < 0.0001, F = 15.28, df(treatment) = 3, df(residual) = 22. A multiple comparison test (Tukey) in these showed significant difference when comparing DMSO against citral or LE135 (*p* < 0.001) and tazarotene against citral or LE135 (*p* < 0.01) that is represented by different letters atop of the bars. Arrows signal the boundary between the mesentery and the new intestinal rudiment. *n* = biological replicates.

**Figure 12 biomolecules-09-00873-f012:**
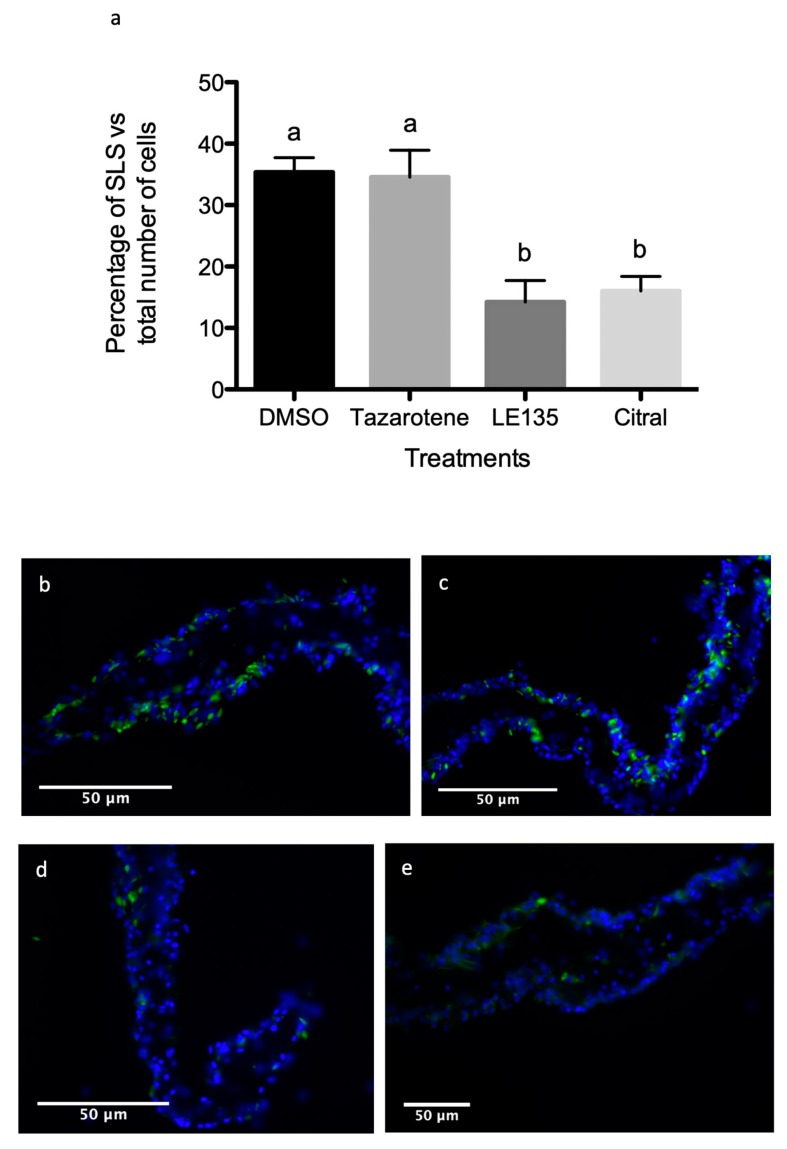
Formation of spindle-like structures by muscle cells during intestinal regeneration. (**a**) The percentage of SLS along the mesenteries near the blastema was established form animals treated with (**b**) DMSO—vehicle, (**c**) tazarotene—RAR agonist, (**d**) citral—RALDH inhibitor, or (**e**) LE135—RAR antagonist following 6 days post evisceration. Between three and five technical replicates and at least three biological replicates were performed per group: DMSO *n* = 9, tazarotene *n* = 4, citral *n* = 10, LE135 *n* = 3. Ordinary one-way ANOVA showed differences among means statistically significant with a *p* value < 0.0001 F = 29.55, df(treatment) = 3, df(residual) = 22. A multiple comparison test (Tukey) in these showed significant difference when comparing DMSO against citral or LE135 (*p* < 0.001) and tazarotene against citral or LE135 (*p* < 0.01) that is represented by different letters atop of the bars. *n* = biological replicates.

**Table 1 biomolecules-09-00873-t001:** PCR primers employed in the current study. Primers labeled with an asterisk (*) were used during the semi-quantitative PCR analyses.

**Short-Chain Dehydrogenase Reductase 7**		
**Primer Name**	**Sequence (5′-3′)**	**Amplicon Product Region**
SDR 5′F	CTCTAGTCAGTAGGCAGAAGGA	5′UTR/ORF
SDR 5′R	TCAGAGAGCACTATCAGGAAGG	5′UTR/ORF
SDR F*	TGACTTTGCTGGTGTTTCCACTGC	ORF
SDR R	CACTGCTCATGTTAGCTATGTGGC	ORF
SDR F2	GCCACATAGTCAACATGAGCAG	ORF
SDR R2*	GCAAGATAGAACTGGGTGAGCA	ORF
SDR 3′R	GCCTATGGATATGCCACTCT	ORF/3′UTR
**Aldehyde Dehydrogenase Family 8A1**		
**Primer Name**	**Sequence (5′-3′)**	**Amplicon Product Region**
ALDH81 5′F	CACTGTAATCTAGAGATTGGTATGGAC	5′UTR/ORF
ALDH81 5′R	GTTTGTGCTGGATGGCAGTTGC	5′UTR/ORF
ALDH81 FA	GCAACTGCCATCCAGCACAAACTA	ORF
ALDH81 RA	AACTTGATCGTACCGTGGTTGGGA	ORF
ALDH81 FB	CGGTACGATCAAGTTTCTCAAATCAGGGG	ORF
ALDH81 RB	GTTTGTTGTGCTCTCTGTTGGAGGTC	ORF
ALDH81 FC*	GACCTCCAACAGAGAGCACAACAAAC	ORF
ALDH81 RC*	GCTGCCCTCTCCACACTCTATTTTC	ORF
ALDH81 FRT2	AGCTGCCTACGAGAATGTGAGCAT	ORF
ALDH81 RRT2	TCTTCCCTTACTGCACCTCCACAA	ORF
ALDH81 FF	GTGGAGGTGCAGTAAGGGAAGA	ORF
ALDH81 3′F	GCATCCTCTAAACACTACCAGTC	ORF/3′UTR
ALDH81 3′R	GATCAAGGGTTCTGTTGATTACTT	ORF/3′UTR
**NADH**		
**Primer Name**	**Sequence (5′-3′)**	**Amplicon Product Region**
NADH F*	CGCAGAAGTAGCCGCGAATAT	ORF
NADH R*	CAATGGTTGTTGCTGGAGTCTTT	ORF

**Table 2 biomolecules-09-00873-t002:** The overall identity/similarity (Blosum62 with threshold 1) of the deduced amino acid sequence of *Holothuria glaberrima* short-chain dehydrogenase reductase 7 or aldehyde dehydrogenase family 8 A1 when compared with the sequences from other organisms. All values are presented in percentage.

Short-Chain Dehydrogenase Reductase 7			Aldehyde Dehydrogenase Family 8 A1	
Organism’s Nickname	Identity (Similarity )		Organism’s Nickname	Identity (Similarity)
scallop	41 (59)		hydra	55 (66)
bristle worm	39 (59)		sea slug	54 (66)
oyster	41 (57)		oyster	57 (69)
drosophila	27 (45)		horseshoe crab	50 (65)
apis	26 (42)		spider	45 (62)
urchin	46 (70)		starfish	55 (69)
lancelet	40 (58)		urchin	44 (60)
ciona	22 (36)		urochord	45 (59)
frog	37 (54)		sea squirt	52 (62)
mus	36 (53)		lancelet	57 (69)
homo	37 (53)		frog	56 (70)
			mus	57 (72)
			homo	57 (71)

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
