# Peer review of "Retinoic Acid Signaling Is Associated with Cell Proliferation, Muscle Cell Dedifferentiation, and Overall Rudiment Size during Intestinal Regeneration in the Sea Cucumber, Holothuria glaberrima"

_biomolecules, 2019, doi:10.3390/biom9120873_

Round 1
Reviewer 1 Report
In this paper, the authors identify and characterize two putative enzymes involved in retinaldehyde and RA synthesis (SDR7 and ALDH8A1) in the sea cucumber. They also demonstrate that treatment of eviscerated cucumbers with either citral or LE135, reduces the level of intestinal regeneration. This manuscript represents a substantial body of relevant and interesting research that adds to the growing literature describing the diverse organisms possibly utilizing retinoid signaling during regeneration of various organs. My main concerns with this manuscript are the semi-quantitative nature of the q-PCR analysis of transcript levels and the clarity of the manuscript. In addition to grammatical errors, there are a number of scientific content errors e.g. in the tables (primers) and images (phylogram) that need to be corrected.
General comments:
I believe it would greatly enhance the clarity of this paper if the process of intestinal regeneration in this organism was more clearly explained and much earlier in the manuscript (i.e in the Introduction), as opposed to in the concluding paragraph, where details of the timing of the intestinal regeneration process were finally provided. This information should be elaborated on in the Introduction so that the timing of tissue harvesting (eg. 3 to 14 dpe) can be placed into context by the reader. Further background on the evisceration process would also be helpful. How much of the intestine is removed? Do all sections of the intestine regenerate (i.e anterior vs medial vs posterior)? Knowing this is important to understanding why transcript levels were measured in these areas.
Does the sea cucumber itself initiate this evisceration process (in response to KCl), and if so, why? Or is this process performed by the experimenter? These details (for readers not familiar with this organism) would be helpful. Likewise, for those not familiar with echinoderms, it would be helpful to briefly define what “ambulacral” deuterostomes are (lines 256-267).
I’m not really sure that Table 1 is particularly helpful and/or contains important information, and could perhaps be either condensed in format (single spacing) or placed as an appendix or a supplementary figure?
Also – considering that LE135 is a selective RARβ antagonist, it would be helpful to be told in the Introduction whether this species actually possesses a RARβ isoform.
The authors seem to have mislabeled the taxonomic groups in Figure 6, where acorn worm, urchin and sea cucumber are labeled in green as lophotrochozoans, and bristle worm, scallop and oysters are labeled in red as deuterostomes. This should be the other way around (though they are correct in fig.7).
Table 2: Why are the primers for ALDH8A1 named RXR? The authors might want to check whether these are just typos (for all 13 primers) or whether primers for the RXR have actually been copied into this table in error.
Full ANOVA results (F values and dfs) should be reported for each data set.
The semi-quantitative analysis of transcript levels initially reported in section 3.3 are very unclear. How does Fig. 8 reflect significant increases in non-regenerating regions of the intestine? Looking at the control bars in each of the 3 graphs in Figure 8, I cannot determine how the medial and posterior segments show a significant increase in expression of the SDR7 transcript (and compared to what? – the anterior region? – this is not specified). This description of these results should be clearer, and perhaps depicted in a separate graph in figure 8 if need be.
Semi-quantitative PCR is clearly not as reliable a method for detecting subtle changes in transcript levels as quantitative PCR, especially using only one normalizing gene. As such, statements in the discussion about such changes should be toned down. E.g. SD7 ortholog is likely differentially expressed” (line 437)
The authors should also discuss the fact that the analysis was only semi-quantitative in nature and that the increases they found in SD7 in the medial and posterior portions were only significantly increased compared to 3dpe (where the levels were non-significantly reduced), rather than to control levels, and what this may mean. The authors should also point out in the discussion that changes in transcript levels do not necessarily mean correlative changes in protein levels.
The first paragraph of the discussion, was mainly repetitive of the information provided in the results section. Here, it would have been far more helpful to explain the importance of some of these regions of the SD7 molecule - such as the relevance or function of the Rossman-fold?
It would be a good idea to explain what changes occur in the retinoid receptor transcripts at various time points of regeneration and how these might relate to the current findings. The table (table 4) somewhat addresses this, but the results are also somewhat confusing to interpret in this table format as not all levels are being compared to non-regenerating control levels.
There is no indication as to whether this organism contains RA. The authors suggest looking at the intestine, but is RA found elsewhere in this organism? This information should be included.
The conclusion paragraph is very long-winded and contains entirely new information (including important info about the timing of regeneration which should have been in the introduction). I suggest the authors substantially reduce their conclusion statements.
Minor comments:
In the Introduction, citations are provided for regeneration of zebrafish and Xenopus (line 66), but no citations are provided for urodele amphibians (line 88). These should be included.
The authors need to perform a thorough grammatical proofread of the manuscript before it is published. In the abstract, there are grammatical errors on lines: 14 (“being the intestinal system the first one to regenerate”, 23 (should read significant differences); and on line 26, the writing suggests that citral is the enzyme synthesizing RA rather than the inhibitor for that enzyme.
Other grammatical / typographical errors are found on lines: 115 (thru), 117 (Genbank), 180 and 183 (divided by - not against); 212 (alpha not alfa); 231 (incomplete phrase), 523 (have), to name but a few.
“here perfomed” and “here presented” should also be re-written throughout.
The authors are also very inconsistent with their use of Italics for species names (eg. Xenopus and H. glaberrima – eg, line 220). Furthermore, the common name for Aplysia is not “snail slug” but “sea slug”.
In Table 3: Xenopus is not a “common name” as listed in the right-hand column.
The quality of the nucleotide and amino acid figures (Figures 1 and 4) are poor (blurry) and I highly recommend higher quality figures are made or used.
The concentration of DMSO used should be stated. Was this equivalent to the highest concentration used for the various drugs?
A brief mention in the methods section that phalloidin labels actin would be helpful.
What do “L” “S” and ‘F” associated with RAR nomenclature refer to? I suspect L and S are long and short forms, but I am not familiar with F.
The concentration of LE135 used in the urodele amphibian studies was not 10µM, but 1 µM.
I thus recommend the authors also check concentrations quoted from other papers in this section of the discussion.
In Fig 10 legend, are the technical replicates different slices from the same animal? And do the n values given represent the biological replicates? Please make this clearer.
Also – in legends 10, 11, and 12 – what do the *** in brackets represent? There are no *** shown on the graphs.
Reviewer 2 Report
The authors report the effect of RA on intestinal regeration of sea cucumbers. They detected expression of SDR7 and ALDH8A1 in the regerating and normal intestine. They also report the effects of drug treatments.
First part of the paper is devoted for molecular characterization, which I do not think properly presented. Most of the data do not have to be presented as they are. I suggest the authors to select truely essential data to be presented in the manuscript. Phylogenetic analyses were not properly done. I suggest the authors to choose the sequences to be included in their analyses. The sequence should be labeled in more understandable manner.
I was not quite persuaded that the difference of gene expression level has significant effect on the concentration of RA. The expression level is already quite high in the normal intestine. Thus, I wonder why the authors think that RA has specific effect on regeneration.
I wonder how the authors exclude the possibility that the drugs such as LE135 or citral has more general toxicity to the sea cucumber.
Reviewer 3 Report
The manuscript "Retinoic acid signalling is associated to cell proliferation, muscle cell dedifferentiation, and overall rudiment size during intestinal regeneration in the sea cucumber, Holothuria glaberrima", written by Viera-Vera J and Garcia-Arraras E. J. presents, in the first part, detection and structural and phylogenic analysis of two holothurian cDNA sequences and their products, respectively, SDR7 and ALDH8A1, belonging to retinoic acid signalling pathway and its metabolism. In the second part of the manuscript, their roles in intestinal regeneration are explored.
The article is well and clearly written. In the Introduction, the metabolism and signalling of retinoic acid are described, as well as the processes of regeneration following evisceration in sea cucumber. In Materials and Methods, methods are clearly presented. Describing sequence analysis, Table 1 is added with GenBank accession numbers. This table, as well as the list of primers used could be put in Supplementary files. Results are well presented. In the paragraph 3.4. BrdU labelling is mentioned in the text, but probably for analysis described in paragraph 3.5, so it would be better to describe its application in paragraph 3.5. Discussion could be shortened for details which are not important in the given context (such as treatment concentrations, precise result numbers from different experiments, use of specific inhibitors if the point is the inhibition of certain signalling molecule, etc.) Conclusion is too long and is written as a continuation of discussion. It should stress the main points from the manuscript. Considering Table 4., it summarizes the results of several investigations, so they should be cited.
I would suggest changing the title of the manuscript, as it covers only one part of the topic, the other is characterisation of ALDH8A1 and SDR7 sequences.
Minor
lane 115: through instead of thru
lane 128: citation in the form of number
lane 137: web page address
from 145 lane on, the whole manuscript: all units should be written separately from the numbers
lanes 165 and 176: 0.1 PBS (0.1x PBS?)
lane 229: Figure 3: cite the program with which the model was done
lane 231: the sequence is 1,212 pb long...
lane 399: Here we have identified the sequences
lane 413: were instead of here
lane 441 and 491: in this model or in the same model
lane 496: Xenopus laevis
lane 503: overactivation
lane 512: ..and redifferentiate helping in the formation of the new intestinal rudiment
lane 542: process
lane 552: expression of the two sequences..
lane 556: Cyp26, an enzyme... has been identified
lane 559: regenerative events of the intestine, to better understand...
lane 569: of these components in our model system, we will...
Round 2
Reviewer 1 Report
The authors have adequately addressed most of my concerns. The most worrisome issue remaining is that most of their conclusions are based on the drug effects of citral (inhibits RA synthesis) and a selective RARß antagonist (LE135), yet according to the author responses, they have not yet shown that this organism actually contains RA and that no RARß isoform is present. Thus, it is not entirely convincing that citral and LE135 are actually exerting selective actions in this organism to perturb RA signaling. I appreciate that the authors are developing the sea cucumber as a potential model for studying RA signaling, which often relies on pharmacological tools, but at the very least, I think the authors need to comment on these issues further in the discussion. Simply providing evidence that other closely related species possess RA; or that citral and particularly the ß-subtype antagonist LE135 have been shown to effect RA-mediated phenomenon in other closely related species, may provide more convincing support for their conclusions. This is important considering they are claiming in their discussion (eg. Line 803), that RA signaling is being “hindered or inhibited”. Use of a RAR-pan antagonist (rather than a selective ß one) may have been advisable. Have they shown that RA itself can affect the animal?
Remaining minor issues related to initial review:
Though the authors dealt with my comment about Table 1 by removing it, I am surprised they removed Table 2 which contained the list of primers used in the study. I would have deemed this important information to retain in the methods section.
Fig 6 and 7 legends refer to Table 2 for containing sequence information used in analysis. I believe this info is in Table 1?
Although Figs 10 and 11 now contain the ANOVA data, Figs 8 and 9 do not.
I think it would be sufficient to express the ANOVA and df data in the typical manner
(eg. Line 462: F (3, 22) = 15.87).
Ref 16 has “51” associated with it.
Author Response
Remaining minor issues related to initial review:
Though the authors dealt with my comment about Table 1 by removing it, I am surprised they removed Table 2 which contained the list of primers used in the study. I would have deemed this important information to retain in the methods section.
Table 1 and Table 2 were moved to the supplementary material in our first revision. Table 1 is still in the supplement and within the text defined as Supplementary Table 1. Table 2 was moved back to the main text and is now Table 1.
Fig 6 and 7 legends refer to Table 2 for containing sequence information used in analysis. I believe this info is in Table 1?
Done
Although Figs 10 and 11 now contain the ANOVA data, Figs 8 and 9 do not. I think it would be sufficient to express the ANOVA and df data in the typical manner (eg. Line 462: F (3, 22) = 15.87).
ANOVA data was added to figure 8 and Figure 9.
Ref 16 has “51” associated with it.
“51” was deleted.
Reviewer 2 Report
Most of my comments were addressed.
Author Response
Most of my comments were addressed.
DONE
Reviewer 3 Report
The authors accepted the comments considering their article.
Minor comments: line 187: when mentioning the addition of address I was thinking on the addition of internet address (such as https://eu.idtdna.com/pages), not the word address.
there are still some measuring units written attached to the numbers (lines 799 and 811)
line 806: a RA...
Author Response
The authors accepted the comments considering their article.
Minor comments: line 187: when mentioning the addition of address I was thinking on the addition of internet address (such as https://eu.idtdna.com/pages), not the word address.
Done
there are still some measuring units written attached to the numbers (lines 799 and 811)
Done
line 806: a RA...
Done